# Variant Interpretation in Current Pharmacogenetic Testing

**DOI:** 10.3390/jpm10040204

**Published:** 2020-10-31

**Authors:** Sally Luvsantseren, Michelle Whirl-Carrillo, Katrin Sangkuhl, Nancy Shin, Alice Wen, Philip Empey, Benish Alam, Sean David, Henry M. Dunnenberger, Lori Orlando, Russ Altman, Latha Palaniappan

**Affiliations:** 1Department of Pharmacy, Stanford Health Care, Stanford, CA 94305, USA; sluvsantseren@gmail.com (S.L.); nshin@stanfordhealthcare.org (N.S.); awen@stanfordhealthcare.org (A.W.); 2Department of Biomedical Data Science, Stanford University, Stanford, CA 94305, USA; mwcarrillo@pharmgkb.org (M.W.-C.); katrin@pharmgkb.org (K.S.); russ.altman@stanford.edu (R.A.); 3Department of Pharmacy and Therapeutics, Center for Clinical Pharmaceutical Sciences, University of Pittsburgh School of Pharmacy, Pittsburgh, PA 15261, USA; pempey@pitt.edu; 4Department of Pharmacotherapy & Translational Research, University of Florida College of Pharmacy, Gainesville, FL 32610, USA; b.alam@cop.ufl.edu; 5Department of Family Medicine, University of Chicago Pritzker School of Medicine, Chicago, IL 60637, USA; spdavid1@uchicago.edu; 6Mark R. Neaman Center for Personalized Medicine, NorthShore University Health System, Evanston, IL 60201, USA; MDunnenberger@northshore.org; 7Department of Medicine and The Center for Applied Genomics and Precision Medicine, Duke University, Durham, NC 27708, USA; orlan002@duke.edu; 8Division of Primary Care and Population Health, School of Medicine, Stanford University, Stanford, CA 94305, USA

**Keywords:** pharmacogenetics, pharmacogenomics, CPIC, precision medicine, nomenclature, standardization

## Abstract

In the current marketplace, there are now more than a dozen commercial companies providing pharmacogenetic tests. Each company varies in the panel of genes they test and the variants they are able to screen for. The reports generated by these companies provide phenotypic interpretations of pharmacogenes and clinically actionable gene–drug interactions based on internally curated data and proprietary algorithms. The freedom to choose the types of evidence to include versus exclude in interpreting genomics has created reporting discrepancies in the industry. The case report presented here reveals the discordant phenotype analysis provided by two pharmacogenetic testing companies. The uncertainty and unnecessary distress experienced by the patient highlights the need for consensus in phenotype reporting within the industry.

## 1. Introduction

A 65-year-old treatment-naïve Caucasian woman was seen in a Pharmacogenomics Consult Clinic for advice on anxiety and depression medications after learning from a genetic test through Company A that she is a Cytochrome P450 2D6 (CYP2D6)-poor metabolizer. The genetic report recommended the patient be cautious of the entire class of selective serotonin reuptake inhibitors (SSRIs) due to “moderate to significant gene-drug interactions”. The patient was concerned as the CYP2D6-poor metabolizer status may affect anti-depressant selection, effectiveness, and tolerability. The patient also reported that she was advised to avoid codeine by her pharmacist, given her CYP2D6-poor metabolizer status. The patient later elected to receive a second genetic test through Company B, which classified her CYP2D6-predicted phenotype as an intermediate metabolizer. Though the CYP2D6 genotypes reported by Company A and Company B were the same for each variant, the phenotypic translation differed in the reports. The two reports also disagreed on CYP2C19 and CYP1A2 phenotypes (see Table 1). The lack of agreement between the two reports worsened the patient’s anxiety surrounding medication response and tolerability presently and in the future. This case highlights the current lack of consensus in pharmacogenetic variant interpretation in the marketplace, and how this can adversely affect patient and provider clinical decision making.

## 2. Discussion 

Genetic tests offered by Company A and Company B are ordered by a provider and buccal swab samples are sent to the respective labs for analysis. Both companies generate a report including genotypes, phenotypes, and result interpretation statements regarding each phenotype, including possible gene–drug interactions. The genetic tests marketed by Company A and Company B are lab developed tests and were not reviewed by the U.S. Food and Drug Administration (FDA). In 2018, the FDA warned the public regarding genetic tests with claims to predict response to specific medications and that changing treatment based on interpretations made by individual companies could lead to inappropriate treatment decisions and potentially have serious health consequences for patients. The FDA recognizes that the use of some drugs can be informed by pharmacogenetic testing (i.e., clopidogrel and *CYP2C19*) and, beyond the drug labelling itself, now provides a Table of Pharmacogenetic Associations (https://www.fda.gov/medical-devices/precision-medicine/table-pharmacogenetic-associations), for which data support therapeutic management recommendations. The FDA recommends that providers make their individual assessment of genetic test results and weigh the strength of evidence when making prescribing decisions.

Both Company A and Company B are considered reputable in the industry and stand by their interpretations of the tests, however the evidence supporting their claims is not easily accessible and the thought process is not transparent. For this case study, we referred to published guidance by Clinical Pharmacogenetics Implementation Consortium (CPIC). The CPIC is an international consortium of volunteers dedicated to writing evidence-based drug-gene guidelines for the use of pharmacogenetic test results in clinical practice. The CPIC guidelines are peer-reviewed, open-access, and many are endorsed by professional societies such as the American Society of Health System Pharmacists. The organization’s transparent processes inspire clinician confidence and is increasingly used as the basis for clinical decision support in pharmacogenomic implementations.

In the case of our patient, both companies reported the same genotypes for *CYP2D6* and *CYP2C19* but disagreed in their phenotypic classifications. For the CYP1A2 enzyme, the companies tested for different variants, and therefore reported different genotypes, as noted in Table 1. The significant reporting discrepancy between the two companies are analyzed and compared against the CPIC guidance in this report.

*CYP2D6* is a highly polymorphic gene involved in the metabolism of 20% of medications, notably SSRIs, tricyclic antidepressants (TCAs), codeine, tramadol, and tamoxifen [1,2]. The metabolic status of *CYP2D6* is determined by the sum of activity scores assigned to each allele. The historical inconsistency in the phenotypic classification of CYP2D6 was due to variable activity score cutoffs across laboratories. In August 2019, the CPIC published a consensus recommendation for the standardization of *CYP2D6* genotype to phenotype translation. In this guidance, *CYP2D6*4/*41* is assigned a total activity score of 0.5, which corresponds to a predicted phenotype of intermediate metabolizer which did not differ from previous CPIC guidance. An intermediate metabolizer with an activity score of 0.5 corresponds with 30–40% CYP2D6 metabolic capacity, while a poor metabolizer has an activity score of 0, which corresponds to 0–10% CYP2D6 metabolic capacity [2]. Company A classified the patient as a poor metabolizer and Company B classified the patient as an intermediate metabolizer; such discrepancies in phenotypic classification of *CYP2D6*4/*41* can impact clinical decision making. For example, there would be different therapeutic recommendations regarding paroxetine, an SSRI used in depression. The CPIC recommends selecting an alternative drug therapy not predominantly metabolized by CYP2D6 or a 50% paroxetine dose reduction when initiated in patients with a CYP2D6-poor metabolizer status while patients with an intermediate metabolizer status can be initiated on the standard starting dose [1]. Clinically, this can impact a patient’s time to response to therapy and remission. The patient in our case revealed her CYP2D6-poor metabolizer status to her pharmacist, as reported by Company A, and received advice to avoid codeine, an analgesic medication that requires a functional CYP2D6 enzyme for its conversion to morphine. While the advice was sound, the pharmacist was not informed of the patient’s *CYP2D6* genotype and variation in the phenotypic classification between different laboratories. The CPIC *CYP2D6*-codeine guideline does not explicitly recommend against the use of codeine use in intermediate metabolizers as it does for poor metabolizers. In our case study, Company B’s phenotypic classification of *CYP2D6 *4/*41* as an intermediate metabolizer is in concordance with the CPIC guidance, and therefore should ultimately be used to provide counseling and clinical guidance.

As with *CYP2D6*, *CYP2C19* is a highly polymorphic gene responsible for metabolizing 5–10% of medications, including SSRIs, TCAs, and clopidogrel [3]. In February 2017, the CPIC published guidance on the standardization of the *CYP2C19* allele’s functional status and phenotype [4]. Per this guidance, the phenotype of CYP2C19 is inferred from its two alleles. Our patient carries the **1* and **17* alleles, which have a functional status of normal function and increased function, respectively. Together, the combination of these two alleles predicts a rapid metabolizer phenotype. In the 2017 CPIC term standardization consensus, the term “extensive metabolizer” was replaced with “normal metabolizer” to reduce confusion amongst clinicians. Normal metabolizers have fully functional enzyme activity (combinations of normal function and decreased function alleles) [3]. In our case study, Company A classified the patient as an extensive (normal) metabolizer and Company B classified the patient as a rapid metabolizer. As with CYP2D6, this discrepancy leads to different therapeutic recommendations. For example, the SSRI escitalopram is a commonly used first-line antidepressant. CYP2C19-rapid metabolizers are predicted have a lower serum concentration of escitalopram, increasing the likelihood of treatment failure. The CPIC recommends an alternative antidepressant in these patients, whereas CYP2C19-normal metabolizers are expected to have appropriate serum levels of escitalopram and a greater likelihood of responding to treatment [1]. The disagreement between the two reports created more uncertainty for this patient and the clinicians with respect to treatment selection, and ultimately the decision was made to follow Company B’s CYP2C19 phenotype classification, as it was in agreement with the CPIC guidance.

CYP1A2 is an enzyme exclusively expressed in the liver, responsible for the elimination of 8–10% of clinically relevant medications and for many drugs CYP1A2 is not the sole metabolizing enzyme. Genetic variation determines up to 42% of the enzyme activity and the remaining activity is determined by epigenetic and environmental factors [5,6]. Company A reported finding two single-nucleotide polymorphisms (SNPs). These SNPs are known have low linkage disequilibrium, but can also be found together on the same chromosome (https://www.pharmvar.org/gene/CYP1A2). The 5347 C/T variant (rs2470890) is a synonymous amino acid SNP, and when seen independently from other CYP1A2 variants, defines the *CYP1A2*1B* allele. The -163 C/A variant (rs762551) is upstream of the gene’s start codon, and when seen independently from other CYP1A2 variants, defines the *CYP1A2*1F* allele. These two variants are seen together in multiple *CYP1A2* alleles, including **17* and **21*. There is no information in Company A’s report as to if these variants were found on the same or opposite chromosomes, so it is not possible to know which specific alleles were found. According to Company B’s report, it does not assay alleles that include the 5347 C/T variant, therefore that variant was not reported by Company B. Instead, Company B reported only the **1F* allele along with the reference allele **1A*. Typically, synonymous variants, such as **1B*, or 5347 C/T, are not considered to have a phenotypic impact and would have similar functionality to the reference allele **1A*. Therefore, only the **1F*, or −163 C/A, variant would theoretically affect patient phenotype [6]. Currently, there are no drugs that have FDA or CPIC guidance for the use of *CYP1A2* genotypic and phenotypic interpretation to guide therapy. The Dutch Pharmacogenetics Working Group (DPWG) states that both **1A/*1F* and **1B/*1F* are considered extensive metabolizers (i.e., normal metabolizers) and have increased inducibility [7]. *CYP1A2*1F* is only inducible in the presence of CYP1A2 inducers, which include hydrocarbons found in cigarette smoke, omeprazole, carbamazepine, modafinil, char-grilled meats, and brassica vegetables such as broccoli and brussels sprouts [6]. As the causes of variations in CYP1A2 activity can be largely non-genetic, phenotyping of CYP1A2 is less useful in clinical decision-making but could be valuable in understanding certain drug interactions [7]. In our case study, Company A classified the patient genotype based on its functional status and Company B classified the genotype by its inducibility potential.

Based on the above information, the clinician opted to use the phenotypic interpretation provided by Company B for *CYP2C19* and *CYP2D6* as it matched the CPIC guidelines for phenotype interpretations. Our patient was recommended to consider treatment with paroxetine, fluvoxamine, or sertraline at standard initial dose and titrate based on response, and recommended against first-line use of citalopram and escitalopram due to increased probability of treatment failure given her 2C19-rapid metabolizer status. In this scenario, our patient was fortunate to have access to extensive pharmacogenomics consultation services to help understand the discrepancy between the two reports. Further, many consumers are not able to pay out-of-pocket for two genetic tests and receive professional pharmacogenomic consultation. Knowing the complexity of the antidepressant response, our patient hoped to better inform herself and her other providers with pharmacogenetic testing to increase confidence in medication selection. While genomics is just one part of the equation when making treatment selection in depression, it is nonetheless a useful tool that can increase confidence and avoid potential adverse events. We support the use of evidence-based pharmacogenomics in helping to better care for our patients, however, this case clearly illustrates how discordant reporting and interpretations of clinically significant pharmacogenes remains a barrier to optimal care.

In the current marketplace, vendors often perform their own proprietary data curation in making phenotypic assignments and treatment recommendations, which is not readily accessible to clinicians. Although genetic testing companies are under no obligation to adhere to the CPIC guidelines, we see many reasons why convergence with the CPIC guidance as the standard would be ideal across the industry. First, the CPIC engages a large cross-section of scientists from industry, academia, and government, including an international representation which balances and buffers areas of disagreement and conflicts of interest. Second, the CPIC evidence review is transparent, with guidelines that are routinely revised in response to new data, so that any interested party has a forum for proposing changes based on new evidence. Finally, the availability of the CPIC guidelines lessens the burden on practitioners to conduct independent evaluations of quality of the advice provided by each company. Indeed, a label of “fully compliant with the CPIC” would be useful to help clinicians choose genetic testing vendors with confidence. As pharmacogenomic tests become more accessible to consumers, we expect that more patients will encounter genotype, phenotype, and therapeutic recommendation discrepancies. A move towards standardization of variant testing and interpretation within the industry would increase the utility of pharmacogenetic tests and facilitate greater adoption in clinical practice. 

Patient consent was granted for publication of pharmacogenetic results.

## Figures and Tables

**Table 1 jpm-10-00204-t001:** Genotype and phenotype reports from Company A and Company B along with available Clinical Pharmacogenetics Implementation Consortium (CPIC) guidance.

Cytochrome P450 (CYP) Enzymes	Company A		Company B		CPIC Guidance
	Genotype	Phenotype	Genotype	Phenotype	
CYP2D6	**4/*41*	Poor Metabolizer	**4/*41*	Intermediate Metabolizer	**4* = No function
					**41* = Decreased function
					Activity score = 0.5
					Intermediate metabolizer
CYP2C19	**1/*17*	Extensive (Normal) Metabolizer	**1/*17*	Rapid Metabolizer	**1* = Normal function
					**17* = Increased function
					Rapid Metabolizer
CYP1A2	5347C>T-C/T	Extensive (Normal) Metabolizer	**1A/*1F*	Hyperinducer	Guidance unavailable
	−163C>A-C/A				

*: allele.

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
