# Peer review of "Variant Interpretation in Current Pharmacogenetic Testing"

_jpm, 2020, doi:10.3390/jpm10040204_

Round 1

Reviewer 1 Report

The manuscript of Luvsantseren and colleagues reports the case of a patient who receive discordant interpretation of pharmacogenetic results from two commercial kits. The issue described in this case report is paradigmatic about the difficulties to select an appropriate genetic test that leads to a correct phenotype interpretation according to published international guidelines. Therefore, I considered the manuscript interesting and below I have listed some comments and suggestions.  

Was the patient who decided on her own to perform both the pharmacogenetic analyses or they were prescribed by a physician? This is an important issue that can be splitted in two key points. First, the physician should be experienced in prescribing the pharmacogenetics analyses after obtaining the patient’s consent by choosing the most appropriate test based on the list of variant alleles. Second, the physician should interpret the results when the inferred phenotype by the commercial kit is discordant with respect to recognized guidelines (i.e., CPIC), as the Authors pointed out in lines 164-166.

I totally agree with the conclusions of the manuscript but I personally think that a careful education of physicians in prescribing pharmacogenetic analyses and interpreting results will return more rapid results with respect to the harmonization of kits produced by the industry.

Even if it is impossible to identify the patient, I suggest to include a brief statement about the patient’s approval for the publication of her genetic results. 

Author Response

Dear Reviewer,

Thank you for giving us the opportunity to improve our manuscript. We appreciate the time and effort that you have dedicated to providing feedback on our case report. Please see below, in blue, response to the reviewers’ comments and concerns.

Was the patient who decided on her own to perform both the pharmacogenetic analyses or they were prescribed by a physician? This is an important issue that can be splitted in two key points. First, the physician should be experienced in prescribing the pharmacogenetics analyses after obtaining the patient’s consent by choosing the most appropriate test based on the list of variant alleles. Second, the physician should interpret the results when the inferred phenotype by the commercial kit is discordant with respect to recognized guidelines (i.e., CPIC), as the Authors pointed out in lines 164- 166.

In line 150, we disclose that the patient initiated the PGx tests. Although the tests are patient initiated, the order forms are submitted by physicians. In our patient case, the test by Company A was ordered by the patient’s local primary care physician and the second test by Company B was ordered by the physician providing the PGx consult. It is possible that the primary care provider may not have been trained to properly choose and interpret pharmacogenetic results which prompted the patient seek a second opinion through clinic specialized in pharmacogenomics. We agree that ordering physicians should perform an independent review of the reported genotypes using recognized guidelines.  

I totally agree with the conclusions of the manuscript but I personally think that a careful education of physicians in prescribing pharmacogenetic analyses and interpreting results will return more rapid results with respect to the harmonization of kits produced by the industry.

Even if it is impossible to identify the patient, I suggest to include a brief statement about the patient’s approval for the publication of her genetic results.

We agree and will add a statement to the end of our report.

Kind regards,

Sally Luvsantseren, PharmD

Reviewer 2 Report

This short review deals with an issue that in the future, based on an always more increased use of pharmacogenetics tests, could strongly impact on clinical decisions, i.e. different phenotypic interpretation and/or different identification of genetic variants in pharmacogenes. The authors to avoid this risk recommend to follow recommendations of the Clinical Pharmacogenetics Implementation Consortium (CPIC).

The review is very interesting, well written and carefully updated. However, the authors mentioned the Dutch Pharmacogenetics Working Group guidelines limited to the CYP1A2 polymorphisms identified in the patient. They could mention recommendations of the Dutch Pharmacogenetics Working Group also in relation to the CYP2D6 and CYP2C19 patient genotypes.

Author Response

Dear Reviewer,

Thank you for giving us the opportunity to improve our manuscript. We appreciate the time and effort that you have dedicated to providing feedback on our case report. Please see below, in blue, response to the reviewers’ comments and concerns.

The review is very interesting, well written and carefully updated. However, the authors mentioned the Dutch Pharmacogenetics Working Group guidelines limited to the CYP1A2 polymorphisms identified in the patient. They could mention recommendations of the Dutch Pharmacogenetics Working Group also in relation to the CYP2D6 and CYP2C19 patient genotypes.

We referred to the DPWG’s publication on pharmacogenetics background for CYP1A2 gene (https://www.knmp.nl/downloads/g-standaard/farmacogenetica/english-background-information/cyp1a2-english.pdf/view), which gives a great overview of the CYP1A2 gene. Currently, there is no drug-gene DPWG guideline involving CYP1A2.

Kind regards,

Sally Luvsantseren, PharmD